# Heterocellular Adhesion in Cancer Invasion and Metastasis: Interactions between Cancer Cells and Cancer-Associated Fibroblasts

**DOI:** 10.3390/cancers16091636

**Published:** 2024-04-24

**Authors:** Hideki Yamaguchi, Makoto Miyazaki

**Affiliations:** Department of Cancer Cell Research, Sasaki Institute, Sasaki Foundation, 2-2 Kandasurugadai, Chiyoda-ku, Tokyo 101-0062, Japan; m-miyazaki@po.kyoundo.jp

**Keywords:** invasion, metastasis, cancer-associated fibroblast, extracellular matrix, heterocellular adhesion

## Abstract

**Simple Summary:**

Invasion of cancer into surrounding tissues is crucial for it to spread to other parts of the body, a process known as metastasis. The characteristics of cancer cells within tumors are significantly influenced by the tumor microenvironment (TME). Cancer-associated fibroblasts (CAFs) are the primary cellular component of the TME and play a pivotal role in cancer progression, including growth, invasion, metastasis, therapy resistance, and immune suppression. Numerous factors mediating interactions between CAFs and cancer cells have been identified, such as growth factors, cytokines, and extracellular vesicles. Recent studies have highlighted the importance of direct contact between CAFs and cancer cells in facilitating cancer invasion and metastasis to distant organs. This review summarizes recent findings on the molecular and cellular mechanisms underlying this direct heterocellular adhesion, providing insights into how CAFs drive cancer invasion and metastasis.

**Abstract:**

Cancer invasion is a requisite for the most malignant progression of cancer, that is, metastasis. The mechanisms of cancer invasion were originally studied using in vitro cell culture systems, in which cancer cells were cultured using artificial extracellular matrices (ECMs). However, conventional culture systems do not precisely recapitulate in vivo cancer invasion because the phenotypes of cancer cells in tumor tissues are strongly affected by the tumor microenvironment (TME). Cancer-associated fibroblasts (CAFs) are the most abundant cell type in the TME and accelerate cancer progression through invasion, metastasis, therapy resistance, and immune suppression. Thus, the reciprocal interactions between CAFs and cancer cells have been extensively studied, leading to the identification of factors that mediate cellular interactions, such as growth factors, cytokines, and extracellular vesicles. In addition, the importance of direct heterocellular adhesion between cancer cells and CAFs in cancer progression has recently been elucidated. In particular, CAFs are directly associated with cancer cells, allowing them to invade the ECM and metastasize to distant organs. In this review, we summarize the recent progress in understanding the molecular and cellular mechanisms of the direct heterocellular interaction in CAF-led cancer invasion and metastasis, with an emphasis on gastric cancer.

## 1. Introduction

Metastasis is the most life-threatening aspect of cancer, and approximately 90% of patients with cancer die due to metastasis. Metastasis is a multi-step process involving various cellular functions. The initial process of cancer metastasis involves the detachment of cancer cells from primary tumors and their invasion into the surrounding stroma [1,2]. In some cases, such as brain tumors, local invasion of cancer cells can be life-threatening [3]. The invasion of cancer cells is triggered by the activation of cell migration and the degradation of the pericellular extracellular matrix (ECM). Cancer cells acquire invasive abilities via phenotypic changes induced by epithelial-mesenchymal transition (EMT) [4]. Recent studies have revealed that cancer invasion is triggered by both cell-autonomous and non-cell-autonomous mechanisms; the former involves the aberrant activation of driver oncogenes, and the latter involves external stimuli from stromal cells and components [5,6].

## 2. Cancer-Associated Fibroblasts in the Tumor Microenvironment

Tumor tissue is composed of cancer cells and diverse cellular and non-cellular components in the tumor stroma that create tumor-supportive niches. The ecosystem surrounding cancer cells is called the tumor microenvironment (TME) and plays a crucial role in tumor development and progression [7]. The TME consists of various non-cancerous cell types, including fibroblasts, immune cells, endothelial cells, adipocytes, and mesenchymal stem cells. Cancer-associated fibroblasts (CAFs), which are myofibroblasts activated by cancer cells, play a central role in the development of tumor-supportive TME [8,9]. Although CAFs mainly originate from resident stromal fibroblasts, other cell types, such as endothelial cells, adipocytes, pericytes, and mesenchymal stem cells, may also be converted into CAFs upon stimulation [8,10]. 

CAFs secrete diverse cytokines, growth factors, and exosomes, through which they reciprocally interact with tumor cells, as well as other cell types, within the TME [11,12]. These multi-network interactions promote the malignant progression of tumors, including angiogenesis, chronic inflammation, immune suppression, and invasion and metastasis [13,14]. CAFs possess a strong ability to build and remodel the ECM through their contractility and production of ECM components (including collagens and fibronectin) and remodeling enzymes (such as matrix-degrading and -crosslinking enzymes) [15]. This results in the generation of a dense and stiff tumor stroma, which accelerates the proliferation and invasion of cancer cells via mechano-signaling and contributes to chemoresistance by physically restricting drug delivery [15]. In addition to influencing primary tumors, CAFs promote cancer metastasis by secreting soluble factors that create metastatic niches in distant organs. Stromal fibroblasts at metastatic sites support the establishment of metastatic lesions; therefore, they are called metastasis-associated fibroblasts (MAFs) [16].

## 3. CAFs Lead Invasion of Cancer Cells

It had long been believed that cancer cells invade the surrounding matrix by themselves. Therefore, studies on cancer invasion have focused solely on cancer cells. However, CAFs have recently been shown to accelerate the acquisition of the invasive phenotype of cancer cells by secreting pro-invasive factors. Over the last two decades, accumulating evidence has shown that CAFs promote cancer invasion by generating a path within the ECM and guiding cancer cells through indirect and direct interactions (Figure 1). In this review, we mainly focus on the role of direct interactions between cancer cells and CAFs in cancer invasion. Details regarding their indirect interactions have been extensively documented in other review articles [12,13,15,17].

### 3.1. CAFs Generate the Path for Cancer Invasion by Remodeling the ECM

In a landmark study, Gaggioli et al. demonstrated the importance of CAFs in the collective invasion of cancer cells [19]. Using an organotypic co-culture system, they found that squamous cell carcinoma cells retaining epithelial markers invaded ECM only in the presence of CAFs. CAFs invaded the ECM via matrix-metalloproteinase (MMP)-dependent proteolytical remodeling and generated the tracks in ECM via mechanical remodeling that requires actomyosin contractility induced by α5 and α3 integrin/Rho-Rock signaling. Cancer cells follow the tracks behind CAFs dependently on Cdc42/MRCK-mediated actomyosin activity. Notably, cancer cell invasion was not induced by the conditioned medium of CAFs or by separating the two cell types. This study demonstrates that even less-invasive cancer cells can aggressively invade through direct interactions with CAFs. Although this study did not investigate the physical contact between cancer cells and CAFs, the fluorescent and timelapse imaging data clearly showed that leading CAFs were in close proximity to, and most likely in direct contact with, the following cancer cells. Indeed, Labernadie et al. showed that the leading CAFs form heterotypic adhesion with the following cancer cells to facilitate collective invasion [20].

Glentis et al. demonstrated that CAFs promote colon cancer cell invasion through the basement membrane via mechanical remodeling of the ECM [21]. CAFs exert contractile forces and generate gaps within the basement membrane through which cancer cells can invade the stroma. In this case, CAFs seem to indirectly accelerate cancer cell invasion via ECM remodeling, rather than via direct contact with cancer cells.

### 3.2. CAF-Led Invasion during Peritoneal Cancer Metastasis

Diffuse-type gastric carcinoma (DGC) is characterized by rapid infiltrative invasion and frequent peritoneal metastasis. DGC is often associated with a massive growth of fibrous stroma due to the extensive proliferation of CAFs [22]. We have previously reported that DGC cells and CAFs, when co-cultured on 3D ECM gels, are attracted to each other and physically come into contact to form invasive cell foci [23]. This phenomenon was not observed when each cell type was individually cultured or stimulated by conditioned media from other cell types. CAFs are localized at the core of cell foci and invade the 3D ECM, bringing about associated cancer cells. Importantly, the DGC cells showed very low invasive activity in this experimental setting. DGC cells activate actomyosin contraction in CAFs via ROCK-dependent phosphorylation of the myosin light chain, accelerating mechanical ECM remodeling and CAF-led invasion of DGC cells.

During peritoneal metastasis of DGC, CAFs within the peritoneum support the colonization and growth of metastasized cancer cells [24]. We recently reported that DGC cells attach to the mesothelium of the peritoneal membrane as multicellular clusters [25] (Figure 2). DGC cells induce mesothelial-mesenchymal transition (MMT), by which mesothelial cells covering the peritoneal surface are converted into CAFs (or CAF-like cells) [26]. Invadopodia are invasive protrusions formed by cancer cells that degrade the pericellular extracellular matrix by focalizing MMP activity, thereby promoting cancer invasion [27]. Cancer cells trigger invadopodia formation in the peritoneal mesothelial cells by upregulating Tks5, a critical regulator of invadopodia formation [28]. This facilitates cancer cell invasion into the submesothelium, led by mesothelial cells undergoing MMT. Tks5 is not abundantly expressed in DGC cells or CAFs derived from DGC tissue. This is consistent with our observation that DGC cells and CAFs exhibit limited invadopodia formation [23]. Therefore, invadopodia formation may be primarily required during the initiation phase of cancer invasion, likely when mesothelial cells undergoing MMT breach the basement membrane to generate a pathway for cancer invasion. When CAF phenotypes are acquired, they may depend more on mechanical remodeling of the stromal ECM driven by actomyosin contractility to promote cancer cell invasion [23]. 

Ovarian, pancreatic, and colon cancers are also known to frequently cause peritoneal metastasis [29]. Similar to DGC, these cancer cells adhere to the mesothelium as multicellular clusters or spheroids to initiate the metastasis process [25,30,31,32]. Interestingly, ovarian cancer cells show strong contact with CAFs, thereby forming heterocellular spheroids in the abdomen [33]. These heterotypic spheroids have a strong ability to adhere to the mesothelium and form peritoneal metastasis. CAFs within the spheroids support the survival and peritoneal invasion of cancer cells. Analogously to DGC, ovarian cancer cells activate and transform peritoneal mesothelial cells into CAF-like cells, which in turn accelerate cancer invasion by co-invading with cancer cells [34]. The precise role of CAFs in the peritoneal invasion of other types of cancer, including pancreatic and colon cancers, has yet to be extensively studied.

## 4. Molecules Mediating the Heterocellular Cancer Cell–CAF Adhesion and Downstream Signaling

Heterocellular adhesion occurs frequently and plays an important role in development. Conversely, heterocellular adhesion is also involved in pathological processes, such as cancer. As described, CAFs interact with cancer cells not only via soluble factors but also via physical attachment, which is presumably mediated by cell surface molecules [5,35]. Several cell surface molecules have been reported to mediate direct cancer cell–CAF interactions, which are required for cancer invasion and metastasis (Figure 3).

### 4.1. Cadherin

Cadherins are a family of transmembrane proteins that play important roles in cell-cell adhesion by assembling adherence junctions [36]. There are several subtypes of cadherins; E-cadherin is expressed in epithelial cells, whereas N-cadherin is expressed mainly in neuronal cells and some other cell types, including fibroblasts. During EMT, cadherin switches from E-cadherin to N-cadherin in cancer cells, in association with the acquisition of an invasive phenotype. Although cadherins typically exhibit homophilic binding, they exhibit heterophilic interactions under certain circumstances, such as embryonic morphogenesis and pathological conditions, including cancer.

Epithelial cells and fibroblasts in contact form adhesive structures that contain E-cadherin, expressed in epithelial cells, and N-cadherin, expressed in fibroblasts [37]. Therefore, these cadherins may form a heterophilic adherence junction. This system appears to be preserved in the heterocellular adhesion between epithelial carcinoma cells and CAFs during cancer invasion. CAFs promote the collective invasion of cancer cells via intercellular physical force [20]. It is transmitted through the adherence junction formed by E-cadherin on the cancer cell membrane and N-cadherin on the CAF membrane. 

Cadherin-11 is a type II classical cadherin initially discovered in osteoclasts. Cadherin-11 is also specifically expressed in fibroblasts and cancer cells that have undergone EMT. A recent study demonstrated that cadherin-11 mediates the formation of adherence junction between cancer cells and fibroblasts [38]. This heterocellular adhesion is required for breast cancer invasion led by fibroblasts. Furthermore, cadherin-11 is expressed in triple-negative breast cancer cells and its high expression is correlated with poor outcomes.

Cadherin-23 is an atypical cadherin primarily expressed in the inner ear and is a crucial component of stereocilia tip links. Cadherin-23 mediates the heterotypic adhesion of co-cultured breast cancer cells and fibroblasts [39]. Cadherin-23 is upregulated in breast cancer tissues, particularly in the stromal regions surrounding the budding duct, which is the invasive front, of tumors. Thus, Cadherin-23 appears to mediate heterotypic adhesion between invading tumor cells and stromal fibroblasts.

### 4.2. Integrin and ECM

Cancer cells often lose cell-cell adhesion due to the downregulation of E-cadherin via EMT. DGC is caused by the loss of E-cadherin function, as evidenced by the fact that hereditary DGC is caused by germline mutations in E-cadherin. Consequently, solitary poorly differentiated cancer cells exist within the massive fibrous stroma of DGC tissues [24]. Therefore, such cancer cells are assumed to use an alternative approach to contact CAFs, independent of E-cadherin.

As described above, we previously reported that direct adhesion of DGC cells to CAFs derived from DGC tissues promotes cancer invasion and peritoneal metastasis [23,35]. We recently screened monoclonal antibodies (mAbs) raised against cell surface molecules of DGC cells to identify inhibitory mAbs that block cancer cell-CAF heterocellular adhesion [18]. Consequently, several blocking mAbs were successfully selected. Surprisingly, all mAbs were found to recognize integrin α5 complexed with integrin β1 subunit. Blocking integrin α5β1 function or expression in cancer cells, or its ligand fibronectin deposited on the surface of CAFs, abrogated heterocellular adhesion. Moreover, knockout of integrin α5 in cancer cells suppressed CAF-led invasion and peritoneal metastasis. Consistent with our findings, Miyazaki et al. independently identified integrin α5β1 in cancer cells and fibronectin in CAFs as key molecules mediating CAF adhesion of pancreatic, lung, and colon cancer cells [40,41], and they showed that blocking integrin α5β1 or fibronectin hampers CAF-led cancer cell invasion in the 3D co-culture system. Integrin α5 is also required for ovarian cancer cells to form heterotypic spheroids with CAFs [33]. These heterotypic spheroids have a high capacity for peritoneal metastasis, with CAFs supporting the survival and peritoneal invasion of cancer cells. Erdogan et al. reported that fibronectin fibers assembled by CAFs promote CAF-cancer cell association and directional cell migration in prostate cancer cells [42]. This fibronectin matrix organization is mediated by CAF contractility and traction forces, transduced to the ECM via integrin α5β1 expressed in CAFs. Furthermore, they demonstrated that αV integrin in cancer cells is crucial for directional cancer cell migration on CAF matrices. Collectively, these studies demonstrate that fibronectin assembled and deposited on CAFs bridges the direct association between cancer cells and CAFs via integrin α5β1 expressed on both cell types during CAF-led cancer cell invasion (Figure 3).

Hirata et al. reported that CAFs provide pro-invasive and -survival signals to melanoma cells in direct co-culture experiments [43]. CAFs create stiff stroma with fibronectin-rich matrices, conferring BRAF-inhibitor resistance to melanoma cells via activation of integrin β1/FAK/Src signaling. Therefore, fibronectin deposition is crucial, not only for mediating heterocellular adhesion but also for indirectly promoting cancer invasion and chemotherapy resistance. Supporting this notion, a recent study demonstrated that direct cell interactions between pancreatic cancer cells and CAFs enhance stem cell phenotypes, including clonogenic growth, self-renewal, and migratory abilities in cancer cells via integrin β1/FAK signaling [44]. Moreover, pancreatic cancer cells promote the expression of type I collagen in CAFs that activates integrin β1/FAK signaling in cancer cells, indicating that heterocellular adhesion establishes a positive feedback loop that promotes cancer stemness. Ovarian cancer cells activate and convert peritoneal mesothelial cells into CAF-like cells. Yoshihara et al. demonstrated that the converted mesothelial cells overexpress fibronectin on their surface and co-invade with cancer cells [34]. Upon direct interaction with cancer cells, the fibronectin also activates Akt signaling, thereby decreasing platinum sensitivity in cancer cells. 

### 4.3. Eph/Ephrin

Normal cells stop continuously migrating in the same direction when they are in contact with other cells. This process is called the contact inhibition of locomotion (CIL). In contrast, metastatic cancer cells are generally defective in CIL against non-malignant cells, whereas they exhibit CIL when in contact with one another. This feature has been proposed to enhance their invasive and metastatic abilities because cancer cells can dissociate from primary tumors through repulsive interactions and invade the stroma by interacting with stromal cells. 

Defective CIL between cancer cells and fibroblasts is mediated by ephrin-Eph signaling [45]. Eph receptors and their ligands, ephrins, are a family of cell surface proteins that play pivotal roles in various biological and pathological processes, such as embryonic development and cancer [46]. Fibroblasts express high ephrin-B2, which activates EphB3/EphB4 in cancer cells. This leads to the activation of Cdc42 signaling, thereby stimulating cell migration and causing defective CIL. Prostate cancer cells were demonstrated to interact with stromal cells expressing ephrin-B2 in human prostate cancer tissues. Furthermore, CAFs display a higher expression of ephrin-B2 [47]. Collectively, ephrin-B2 and EphB3/4 may not mediate physical adhesion between cancer cells and fibroblasts but probably contribute to the co-invasion of the two cell types by inhibiting CIL after their interaction. 

### 4.4. Other Molecules

Nectins belong to the immunoglobulin superfamily transmembrane proteins and are a family of cell adhesion molecules. Nectins are critical regulators of adherence junctions and are necessary for the formation of strong cell-cell adhesions [48]. Nectins and afadin, a protein connecting nectin to the actin cytoskeleton, are recruited to the heterocellular contact sites between cancer cells and CAFs formed through the E-cadherin and N-cadherin interactions described previously [20]. Because afadin depletion in CAFs prevents CAF repolarization, the nectin/afadin system likely contributes to CAF-led cancer cell invasion.

In ductal carcinoma in situ, the presence of PDGFRα^(low)^/PDGFRβ^(high)^ stromal fibroblasts was associated with an increased risk of recurrence. This fibroblast phenotype is induced by direct contact with cancer cells and is mediated by Jagged1 in cancer cells and Notch2 in fibroblasts [49]. Another study demonstrated that migration and invasion of triple-negative breast cancer cells are accelerated by co-culture with CAFs [50]. This process is dependent on physical contact between the two cell types and requires Notch activation in cancer cells [51]. Notch1 in cancer cells activates p65, leading to an elevated release of CXCL8, a pro-metastatic chemokine. Thus, Notch itself may not be important for physical contact but may play a critical role in the activation of cellular signaling in both cell types, resulting in cancer cell invasion.

Using a genetically engineered mouse model, Richardson et al. demonstrated that vimentin is not required for primary tumor growth but is necessary for tumor invasion and metastasis in lung adenocarcinoma [52]. Vimentin is exclusively expressed in CAFs surrounding the collectively invading tumor cells. They demonstrated that vimentin depletion in CAFs suppresses the invasion of CAF and the CAF-led invasion of cancer cells in a 3D spheroid assay. Although it is unclear whether vimentin is directly involved in heterocellular adhesion, it plays an essential role in CAF-led cancer invasion and metastasis.

Otomo et al. have reported that CAFs enhance the invasion and proliferation of lung cancer cells [53]. They showed that direct contact between cancer cells and CAFs is necessary to enhance cancer invasion. Cancer cells contact fibroblasts with tetraspanin 12 and transduce β-catenin signaling in CAFs, leading to the secretion of CXCL6, which promotes cancer invasion. However, the molecule expressed in cancer cells that binds to tetraspanin 12 on CAFs has not yet been identified.

## 5. Targeting Strategy for the Cancer Cell-CAF Adhesion for Cancer Therapy

In the last few decades, it has been well established that CAFs within the TME play a central role in tumor growth and progression. Accordingly, extensive efforts have been devoted to targeting CAFs for cancer treatment. However, the recent development of single-cell sequencing technology revealed that considerable heterogeneity and functional subpopulations exist in CAFs within tumor tissues [54]. Certain CAF subpopulations have been known to restrict tumorigenesis and tumor progression [8,55]. Therefore, strategies to eliminate CAFs may also impact tumor-suppressive CAFs and induce unwanted tumor progression [56]. Additionally, non-specific targeting of CAFs likely inhibits activated fibroblast function in physiological settings, such as wound healing and tissue regeneration. In this context, targeting heterocellular adhesion between cancer cells and CAFs may have a more specific effect on the tumor-promoting functions of CAFs, resulting in enhanced therapeutic benefits. Targeted therapy against cancer cell–CAF interactions should be combined with existing cytotoxic therapies, such as chemotherapy and radiotherapy.

It remains unclear whether CAFs that form heterocellular adhesion with cancer cells constitute a distinct subpopulation. There are two primary subtypes of CAF: myofibroblastic CAFs (myCAFs), characterized by high expression of alpha-smooth muscle actin and strong contractility; and inflammatory CAFs (iCAFs), which have immune-modulating properties by secreting inflammatory cytokines/chemokines [9]. Öhlund et al. reported that pancreatic stellate cells converted into myCAFs through direct interaction with cancer cells, whereas iCAFs can be induced by indirect co-culture [57]. Intriguingly, myCAFs are closely associated with cancer cells, while iCAFs are typically located distantly in pancreatic tumors. In contrast, other studies showed that iCAFs are enriched in invasive layers and are located around cancer cells in DGC [58,59]. As mentioned, CAFs in direct contact with cancer cells play crucial roles in creating pathways and guiding cancer cells during invasion. CAFs located distantly from cancer cells also contribute to CAF-led invasion by secreting ECM components and chemotactic factors, thereby fostering a pro-invasive tumor microenvironment. Given the strong contractility and mechanical ECM remodeling activity of myCAFs, they may be the main facilitator of cancer cell invasion through direct contact. Conversely, iCAFs, known for secreting pro-invasive molecules, such as MMPs, chemokines, and growth factors, may contribute to cancer invasion through indirect interaction within tumor regions, either distant or adjacent to cancer cells. Further studies are essential to pinpoint the specific CAFs, if they exist, responsible for driving CAF-led invasion of cancer cells.

Integrin-ECM signaling emerges as the most promising pathway among the molecules facilitating cancer cell–CAF adhesion for CAF-led invasion and metastasis. We recently reported that administration of the inhibitory antibodies against integrin α5β1, which abrogate the heterocellular cancer cell–CAF adhesion as mentioned previously, blocks peritoneal metastasis of DGC in a mouse xenograft model [18]. In this model, no obvious adverse effects were observed, with antibodies targeting integrin α5β1. Therefore, such therapeutics may have clinical effects against cancer progression mediated by CAFs. Integrin α5β1 is also expressed in endothelial cells and is crucial for tumor angiogenesis. Consequently, several antibody therapeutics have been developed and tested in clinical trials as anti-tumor angiogenesis therapeutics [60]. However, these trials were discontinued due to disappointing results, likely because such clinical trials are generally designed to evaluate tumor shrinkage. Considering its critical role in CAF-led cancer invasion and metastasis, integrin α5β1 antibody therapeutics may need to be evaluated in different aspects of malignant progression, such as metastatic recurrence and metastasis-free survival.

RGD peptides, well-known integrin inhibitors, competitively inhibit integrin-ECM interactions. RGD peptides block cancer cell–CAF heterocellular adhesion, CAF-led cancer cell invasion, and peritoneal metastasis [18,40,41,61]. Although numerous clinical trials testing RGD peptides as anti-cancer therapeutics have failed, they may still be efficacious against the invasive and metastatic progression of cancers, similar to integrin antibodies.

Because integrin signaling is transduced through the activation of non-receptor tyrosine kinases, Src and FAK, inhibitors of these kinases may potentially block downstream signaling activated by cancer cell–CAF adhesion. Treatment with FAK inhibitors has been shown to overcome CAF-mediated resistance to BRAF inhibitors in melanoma [43]. The stem cell phenotypes of pancreatic cancer cells induced by direct CAF interactions are also repressed by treatment with an FAK inhibitor [44]. In a previous study, we screened an inhibitor library and discovered that the Src inhibitor dasatinib blocked the formation of invasive foci mediated by heterocellular adhesion between DGC and CAFs [23,35]. Moreover, we demonstrated that dasatinib effectively suppressed peritoneal metastasis of DGC in a mouse xenograft model. Histological analysis revealed that dasatinib administration reduced the association between metastasized tumors and CAFs. Thus, FAK and Src inhibitors appear effective in blocking the malignant phenotypes of cancer cells elicited via CAF adhesion. However, such inhibitors may have broader effects on various cellular functions than blocking specific integrin isoforms, increasing the possibility of adverse effects.

## 6. Perspectives

Numerous questions and challenges remain unaddressed in comprehending the molecular mechanisms underlying heterocellular adhesion between cancer cells and CAFs. Such understanding establishes a foundation for devising targeted therapies that disrupt heterocellular adhesion and impede CAF-led cancer invasion and metastasis. 

The primary challenge is the heterogeneous nature of CAFs, which are originated from multiple cell types and divided into subpopulations with distinct functions. Furthermore, intra-tumoral, inter-tumoral, and inter-individual heterogeneity of CAF phenotypes and origins may exist. Therefore, CAFs likely exhibit diverse abilities and modes of interaction with cancer cells, based on the original cell type, tissues, and subpopulations. In this context, it is particularly important to delineate subpopulations of CAFs that directly engage in heterocellular adhesion with cancer cells and those that indirectly influence CAF-led cancer invasion. The recent advancements in high-resolution single-cell spatial transcriptome analysis hold significant promise in elucidating the individual and distinct contributions of various CAF subtypes in cancer invasion.

Moreover, understanding the cancer cell types or phenotypes, including genomic and transcriptomic backgrounds, along with the characteristics of the TME that preferentially exploits CAF-dependent invasion, is crucial. Thus, comprehensive analyses of a range of cancer cells and CAFs concerning heterocellular adhesion and resulting invasion, ideally using clinical samples in association with pathological and clinical data, are imperative for understanding the biology and effectively targeting it in cancer therapy. 

Understanding the distinctions and similarities in heterocellular adhesion between epithelial cells–fibroblasts and cancer cells–CAFs, in both physiological and pathological settings, is equally vital. The knowledge is essential for minimizing the adverse effects of targeting heterocellular adhesions. For clinical translation, a precision medicine approach based on the genomic or transcriptomic information of CAFs and cancer cells may be required. Identifying biomarkers associated with cancer cell–CAF adhesion could carry diagnostic and prognostic implications. The development of small-molecule drugs or other modalities that disrupt heterocellular adhesion holds promise for clinical applications. Although further studies are essential, targeting the heterocellular adhesion between cancer cells and CAFs represents an innovative cancer therapy.

## 7. Conclusions

Cancer cells interact with CAFs through both soluble factors and direct physical contacts. Accumulating evidence has shown that the heterocellular adhesion between cancer cells and CAFs is pivotal for cancer progression. In particular, CAFs adhere to cancer cells and drive cancer invasion and metastasis. Hence, disrupting this heterocellular adhesion may offer a new therapeutic approach for aggressive cancers. Recently, several cell adhesion molecules mediating the heterocellular adhesion have been uncovered. These molecules may be promising targets for the development of a novel cancer therapeutics.

## Figures and Tables

**Figure 1 cancers-16-01636-f001:**
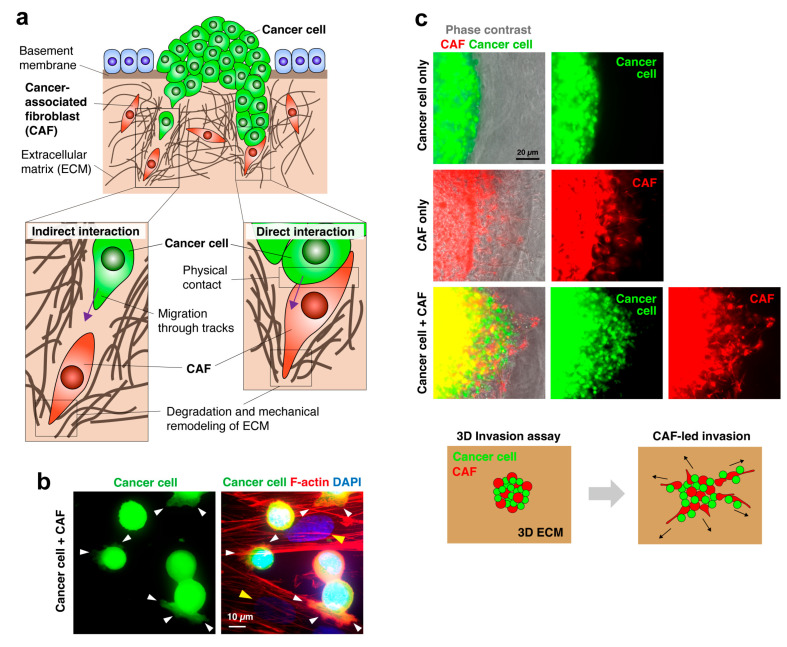
Cancer cell invasion led by CAFs. (**a**) Schematic diagram of CAF-mediated cancer cell invasion. CAFs generate a pathway for cancer invasion through enzymatic degradation and mechanical remodeling of the ECM. Cancer cells either follow the tracks behind CAFs or physically contact and co-invade with CAFs, depending on cancer types and circumstances. (**b**) Fluorescent images of GFP-labeled diffuse-type gastric cancer (DGC) cells adhered to the CAFs (yellow arrowheads). Cancer cells extend actin filament-rich protrusions (white arrowheads) that appear to attach to and grip the CAFs. (**c**) Fluorescent images of the 3D co-culture invasion assay. DGC cells and CAFs were co-embedded within the 3D ECM, and their radial invasion into the surrounding ECM was observed. Cancer cells alone did not invade the ECM, whereas CAFs alone invaded the ECM. When co-cultured, cancer cells can invade the ECM along with CAFs. The lower panel shows the scheme of the invasion assay. Data were reproduced with permission from [18].

**Figure 2 cancers-16-01636-f002:**
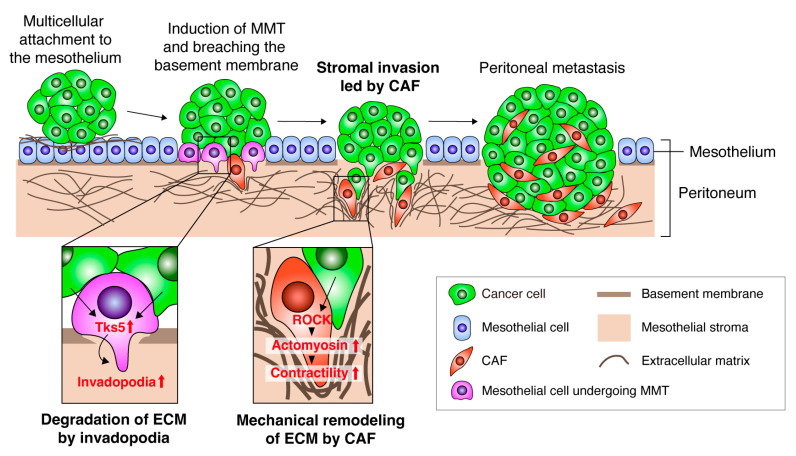
A model of CAF-led cancer cell invasion during peritoneal metastasis. Schematic of the model of peritoneal cancer metastasis. Cancer cells attach to the mesothelium as multi-cellular clusters. Mesothelial cells undergo mesothelial-mesenchymal transition upon stimulation by cancer cells. The converted mesothelial cells degrade the ECM and breach the basement membrane through invadopodia formation induced by the upregulation of Tks5. Cancer cells co-invade with CAFs into the stroma by activating ROCK and actomyosin contractility in CAFs and inducing mechanical ECM remodeling. These processes lead to colonization and growth of metastasized tumors in the peritoneum.

**Figure 3 cancers-16-01636-f003:**
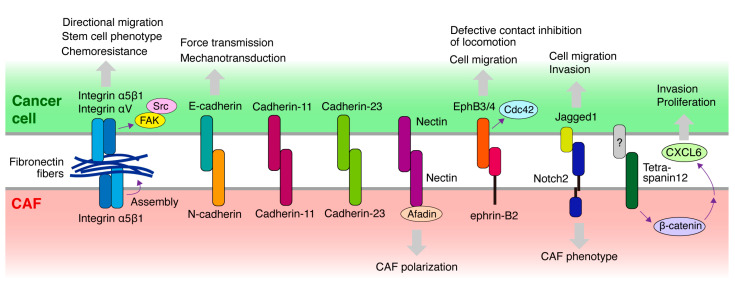
Molecules potentially mediating the heterocellular adhesion or contact between cancer cells and CAFs. Schematic diagram illustrating the molecules expressed in cancer cells and CAFs that mediate heterocellular adhesion or contact between the two cell types and are implicated in CAF-led cancer invasion. Cellular responses elicited by heterocellular adhesion are also shown. Notably, fibronectin fibers assembled and deposited on CAFs bridge the direct association between cancer cells and CAFs via integrins expressed in both cell types.

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
