# Peer review of "Heterocellular Adhesion in Cancer Invasion and Metastasis: Interactions between Cancer Cells and Cancer-Associated Fibroblasts"

_cancers, 2024, doi:10.3390/cancers16091636_

Round 1
Reviewer 1 Report
Comments and Suggestions for Authors
The work discusses the interactions between cancer cells and CAFs. The layout of the work is correct, the illustrations properly refer to the content. The language of the work is correct. It's a good read, but in my opinion, the work is too general. The included information such as integrins, fibronectin, invadopodia, cadherins, and eph is quite obvious and does not cover the latest results on this topic. The authors often refer to scirrhous gastric carcinoma (previously published on this topic), however, the topic also requires further exploration in other types of cancer that occur much more frequently. Targeted therapies are also described very briefly. The literature should mostly cover the last 5 years.
Comments on the Quality of English LanguageMinor editing of English language required
Reviewer 2 Report
Comments and Suggestions for Authors
In their review, Yamaguchi and Miyazaki provide a general introduction on how heterocellular adhesion between cancer cells and CAFs contribute to tumor progression and metastatic dissemination.
The review starts with an introduction on the roles of CAFs in the tumor microenvironment; then it moves on to examine some examples of how CAFs lead cancer cell invasion by direct contact; finally, the authors give an overview of molecules involved in the cancer-cell-CAF adhesion. The idea advanced in the concluding remarks is that targeting CAFs to treat cancer is hard because CAFs are very heterogeneous. Focusing instead on molecules mediating the interaction between cancer cells and CAFs would provide a more targeted approach.
While the concept of targeting more specific aspects of CAFs biology is interesting, the review describes both direct and indirect interactions between CAFs and cancer cells. Moreover, the chapters are too long and do not fully correspond to their title. My suggestion would therefore be to split section 3 (CAFs lead to invasion of cancer cells upon direct contact) in two sections (direct and indirect interactions) and to underscore this distinction in section 4 (molecules mediating the heterocellular cancer cell-CAF adhesion and downstream signaling).
To be more specific:
In section 3, the authors mostly present researches where direct contact between CAFs and cancer cells was not investigated. For example, in Gaggioli et al (which, I agree, is a landmark study), cancer cell invasion is promoted by CAFs generating tunnels in the extracellular matrix. Cancer cells that would not be invasive due to lack of proteolytic activity become then able to move into these pre-generated tunnels. Whether this is due to direct contact between cancer cells and CAFs was not investigated (for example by interfering with cadherins). Similarly, in Glentis et al., it is shown that CAFs render cancer cells invasive by exerting forces on the basal membrane thereby generating holes. The paper from Labernadie et al. (ref. 43), which shows how a direct contact between cancer cells and CAFs promotes invasion, is not even mentioned in this section.
Along the same line, in section 4, a lot of evidence presented is not about direct contact, but mostly indirect contact through remodeling of the extracellular matrix (see for example Refs. Nr. 50 and 51).
Also, Cadherin-11 would probably deserve to be mentioned in the cadherin part (see https://doi.org/10.7554/eLife.87423.1).
Other minor comments:
I appreciated the paragraphs on antibody therapeutics, but as this part is not the center of the review, I found it odd that the authors just cited one single clinical study when saying that several clinical trials focusing on angiogenesis have been designed (line 341). I would rather cite some more studies, or maybe a review, even if primary research deserves to be cited.
Reviewer 3 Report
Comments and Suggestions for Authors
This is a nicely presented review, focusing on adhesion molecules regulating the cancer cell-CAF interactions that promote invasion and metastasis. This is a topic that is often overlooked in TME studies and little is covered by CAF-related review articles. Identifying biomarkers associated with cancer cell–CAF adhesion would add value to the current efforts in targeting CAFs. However, this direct cancer cell-CAF adhesion only represents a relatively small portion of cells in CAF-rich solid tumors. CAFs are highly heterogeneous and exhibit high plasticity, temporally and spatially, as revealed by many single-cell RNA seq analyses in CAF subpopulation studies. It is unclear if the cancer cell-interacting CAFs represent a distinctive CAF subpopulation. Also, the contribution of those non-adhesive associated CAFs should be discussed to broaden the perspective view of this review.
Author Response
This is a nicely presented review, focusing on adhesion molecules regulating the cancer cell-CAF interactions that promote invasion and metastasis. This is a topic that is often overlooked in TME studies and little is covered by CAF-related review articles. Identifying biomarkers associated with cancer cell–CAF adhesion would add value to the current efforts in targeting CAFs.
We truly appreciate your time and effort in reviewing our work and providing valuable comments. We believe that your feedback has significantly enhanced the overall quality of our paper.
However, this direct cancer cell-CAF adhesion only represents a relatively small portion of cells in CAF-rich solid tumors. CAFs are highly heterogeneous and exhibit high plasticity, temporally and spatially, as revealed by many single-cell RNA seq analyses in CAF subpopulation studies. It is unclear if the cancer cell-interacting CAFs represent a distinctive CAF subpopulation. Also, the contribution of those non-adhesive associated CAFs should be discussed to broaden the perspective view of this review.
Thank you for pointing out this important issue. We described and discussed about subpopulations of CAFs and their adhesion with cancer cells and contribution to cancer invasion in Section 5, lines 344-363, and Section 6, lines 410-414, in the revised manuscript. Although we briefly discussed the contribution of indirectly associating CAFs, several recent articles reviewing it were cited according to the suggestion by Reviewer 4.
Reviewer 4 Report
Comments and Suggestions for Authors
In this review by Yamaguchi and Miyazaki entitled “Heterocellular adhesion in cancer invasion and metastasis: interactions between cancer cells and cancer-associated fibroblasts”, the authors provide an overview as to how cancer cells in tumor tissues are strongly affected by the tumor microenvironment (TME) and interactions with cancer-associated fibroblasts (CAFs). The authors then discuss how cells interactions via indirect or
hetero-cellular adhesion between cancer cells and CAFs in cancer progression are orchestrated.
In general, the work presented is of interest and offers some good updates and key information about cellular interactions in the TME. There are however a few points, listed below, which should be considered to strengthen the delivery of the work presented.
· One important point centres around the fact that whilst the authors have described communications between cancer cells and fibroblast taking place in both a direct and indirect manner in their introduction, the majority if not exclusivity of the work presented here relates to direct interactions via cell receptors. Authors should perhaps refer to other recent work revieing the latter non direct form of communications if they wished not to significant increase the content of the manuscript.
· There is little difference in complexity between the simple summary and the abstract. Perhaps authors should aim to simplify further the former one.
· This is perhaps a superficial point but it is important to remind the authors that cancer cells are by definition metastatic and invasive cells, hence their names. Referring to non-invasive cancer cells as done multiple times in the review (e.g. Line 97) is somewhat misleading and inaccurate.
Author Response
In this review by Yamaguchi and Miyazaki entitled “Heterocellular adhesion in cancer invasion and metastasis: interactions between cancer cells and cancer-associated fibroblasts”, the authors provide an overview as to how cancer cells in tumor tissues are strongly affected by the tumor microenvironment (TME) and interactions with cancer-associated fibroblasts (CAFs). The authors then discuss how cells interactions via indirect or hetero-cellular adhesion between cancer cells and CAFs in cancer progression are orchestrated. In general, the work presented is of interest and offers some good updates and key information about cellular interactions in the TME. There are however a few points, listed below, which should be considered to strengthen the delivery of the work presented.
Thank you very much for carefully reading our manuscript and giving us the valuable and constructive comments below.
- One important point centres around the fact that whilst the authors have described communications between cancer cells and fibroblast taking place in both a direct and indirect manner in their introduction, the majority if not exclusivity of the work presented here relates to direct interactions via cell receptors. Authors should perhaps refer to other recent work revieing the latter non direct form of communications if they wished not to significant increase the content of the manuscript.
Thank you for this suggestion. According to this comments, we refer to other recent reviews describing indirect cancer cell-CAF interactions in lines 93-96 in the revised manuscript.
- There is little difference in complexity between the simple summary and the abstract. Perhaps authors should aim to simplify further the former one.
We agree with the reviewer that the simple summary does not significantly differ from the abstract. We revised the simple summary according to this suggestion.
- This is perhaps a superficial point but it is important to remind the authors that cancer cells are by definition metastatic and invasive cells, hence their names. Referring to non-invasive cancer cells as done multiple times in the review (e.g. Line 97) is somewhat misleading and inaccurate.
Thank you for noticing this point. We corrected the words in Line 111 and 119, accordingly.
Round 2
Reviewer 1 Report
Comments and Suggestions for Authors
I'm very sorry, but despite adding a few fragments, the scientific level of the paper has not changed and doesn't correspond to a journal with an IF > 5. The work is very general and refers to quite common information, not going beyond the level of academic books on cytophysiology. I cannot recommend this article in this version.
Comments on the Quality of English LanguageMinor editing of English language required
Author Response
We appreciate the time and effort you have dedicated to evaluating our work. We understand your concerns regarding the scientific level of this paper. However, we have tried to improve the manuscript based on all reviewers' valuable comments received during two rounds of the review process. We believe that this revised manuscript now meet the standards expected for publication in this journal. Additionally, we would like to emphasize that this review focuses on heterocellular adhesion between cancer cells and CAFs, a topic only briefly addressed in other reviews, as noted by Reviewer 3. Therefore, this review will provide updates and new insights into the molecular mechanisms underlaying cell interactions within tumor microenvironment and cancer invasion.